# Coping Resources among Forced Migrants in South Africa: Exploring the Role of Character Strengths in Coping, Adjustment, and Flourishing

**DOI:** 10.3390/ijerph21010050

**Published:** 2023-12-29

**Authors:** Aron Tesfai, Laura E. Captari, Anna Meyer-Weitz, Richard G. Cowden

**Affiliations:** 1Discipline Psychology, School of Applied Human Science, College of Humanities, Howard College Campus, University of KwaZulu-Natal, Durban 4041, South Africa; meyerweitza@ukzn.ac.za; 2The Albert and Jessie Danielsen Institute, Boston University, Boston, MA 02446, USA; lcaptari@bu.edu; 3Human Flourishing Program, Institute for Quantitative Social Science, Harvard University, Cambridge, MA 02138, USA

**Keywords:** forced migration, African refugees, coping resources, character strengths, existential positive psychology

## Abstract

This phenomenological qualitative study explored how forced migrants in South Africa cope with violent, traumatic experiences and precarious resettlement conditions. Data came from a larger empirical project examining migration, psychological distress, and coping. In-depth interviews were conducted with 14 refugees and asylum seekers (*M*_age_ = 30.27, *SD*_age_ = 9.27; male = 71.43%) who migrated from five African countries to Durban, South Africa. Despite overwhelming stressors, participants described pathways to transcend victimhood and hardship through engaging character strengths in ways that promote post-traumatic growth. Qualitative analysis revealed five overarching domains: spirituality and religiousness, love and kindness, hope and optimism, persistence and fortitude, and gratitude and thankfulness. Findings are framed within positive existential psychology and dual-factor understandings of mental health, which attend to both human suffering and flourishing. Limitations, future research directions, and clinical and community implications are discussed, with attention to the role of character strengths in adaptive coping and psychological well-being. The intergenerational transmission of strengths is explored as one potential means of buffering intergenerational trauma impacts and promoting family post-traumatic growth.

## 1. Introduction

It is estimated that forced displacement in Africa represents nearly one-third (30 million) of the global total [1]. Forced migrants experience trauma and violence, including war, combat, imprisonment, rape, torture, and police brutality [2,3] These individuals and families are forced to flee from unbearable systematic human rights violations that compromise their safety, livelihood, mental health, and well-being. While fleeing, many face additional life-threatening experiences, including kidnapping, torture for ransom, lack of basic necessities, compromised conditions in camps, and arduous asylum processes [4,5]. Post-migration, they often encounter institutional and sometimes violent xenophobia, discrimination, exclusion from social services, and employment challenges [6,7], not to mention normative acculturative stresses. In the South African context, evidence suggests that asylees and refugees experience high levels of insecurity and vulnerability compounded by threats, violence, and a precarious socioeconomic status [8,9].

Research has documented an increased risk of mental health conditions among asylees and refugees, including post-traumatic stress disorder (PTSD), depression, anxiety, and psychosomatic disorders [10]. In addition, acculturative stress, a lack of social support, and other post-migration contextual factors may play intervening roles between trauma exposure and negative mental health outcomes [11]. These experiences often necessitate intensive use of coping resources. Asylees and refugees draw from diverse and often communal practices to cultivate resilience, including spirituality, perspective taking, social connection, and the will to live and survive [12]. The appraising and coping process also includes internal resources, such as character strengths and related adaptive emotions [4]. Furthermore, studies have also documented evidence of post-traumatic growth (PTG) among forced migrants [4,13]. PTG entails positive psychological, spiritual, or interpersonal transformation following trauma, such as a greater appreciation for life, deepened relationships, spiritual growth, awareness of new possibilities, and/or a bolstered sense of personal strength [14]. Despite inconsistent evidence concerning the factors associated with PTG in this population, there is growing evidence of PTG among refugees and asylees [15]. This challenges the view of forced migrants as mere victims of traumatic violence and post-migration stressors.

The present study explores how refugees and asylees in Durban, South Africa deal with psychological distress stemming from violent, traumatic experiences and precarious resettlement conditions. To date, little attention has been given to dual-factor understandings of mental health that attend to both human suffering and flourishing [16]—and the interplay between the two—among this population. Shifting from a deficit-based, symptom-focused medical model to a more holistic, capacity-building framework respects the complexity of asylee and refugees’ experiences and is a core tenet of culturally responsive mental healthcare [17]. In this study, we were particularly interested in (1) the explication of internal capacities that may promote PTG, such as the interplay between character strengths (e.g., hope, love) and positive emotions, and (2) how these internal capacities may contribute to a sense of well-being and flourishing despite systemic oppression and structural challenges.

### 1.1. Character Strengths and Existential Positive Psychology

In the last two decades, the positive psychology literature has explicated the importance of attending to human capacities (often referred to as character strengths) that can support mental health, highlighting the diverse and creative ways people may survive and thrive amidst hardship [18]. Character strengths are multidimensional personal characteristics that lead people to desire and pursue ‘the good’ [19] and thus “enhance the capacity to live well” [20]. One commonly used classification system for organizing character strengths was proposed by Peterson and Seligman [21], which situates each character strength within one of six virtues—courage, humanity, justice, wisdom, temperance and transcendence. For example, the strengths of honesty, bravery, persistence, and zest are nested within the virtue of courage. Each character strength is linked with cognitive and affective components that may support adaptive functioning. While some have argued that virtues are universal, multicultural studies suggest that they are understood and embodied uniquely based on cultural group, intersectionality of identities, and social location [17]. Therefore, exploring an emic understanding of virtues and character strengths is critical.

Positive psychology orients researchers and clinicians beyond a primary focus on pathology and encourages attention to human growth and the potential to cultivate healthier communities and institutions [22]. However, early formulations have been critiqued [16] as potentially over-correcting for the medical model by not meaningfully integrating dialectical tensions of (a) positively valenced emotions (e.g., happiness) and character strengths (e.g., hope) with (b) more difficult and painful emotions (e.g., sorrow, anger), leading to potentially reductionistic applications of character strengths that do not necessarily promote well-being, particularly among marginalized populations. For example, Captari et al. [17] described the phenomenon of *virtue bypass* as “when virtue language or behaviors are used in ways counter to flourishing, such as to (a) oppress and subjugate others or (b) repress and deny one’s own emotions and needs” (p. 663).

Critical and multicultural scholars have begun to address this dilemma by (a) attending to both the dark and bright sides of human existence and (b) considering contextual (e.g., culture, religion) and systemic (e.g., oppression, empowerment) factors. This second wave of positive psychology, referred to as existential positive psychology [16], is particularly well suited to capture the experiences of asylees and refugees, which frequently include joyous and hopeful moments, as well as gut-wrenching and horrific ones. Existential perspectives are quite relevant, since complex dilemmas surrounding mortality, freedom, meaning, and connection are inherent in the forced migration experience. Such advancements in positive psychology can help map how this population may be haunted by past traumas and distressed by a precarious livelihood, while also drawing from strengths like hope and gratitude to find their footing in a forever-altered world.

Moreover, as part of a third wave of positive psychology, recent scholarly work has begun to integrate ecological perspectives, orienting researchers and clinicians “beyond the individual person as the primary focus and locus of enquiry, and exploring the manifold sociocultural factors, systems and processes that impact people’s well-being” [18]. This broader perspective is vital to exploring the complexity of suffering and flourishing among marginalized groups. By attending to existential, systemic, and sociocultural factors rather than individual phenomena alone, this study utilizes the latest theorizing within positive psychology to avoid reductionistic and de-contextualized foci [23].

### 1.2. The Present Study

The present study expands on etic (general) understandings of the forced migration experience by attending to African asylees and refugees’ emic (specific and unique) coping resources, which are often understood and embodied within particular cultural and religious contexts. For example, definitions and practices surrounding character strengths might vary significantly from culture to culture [24]. Although etic research helps identify overarching constructs, emic studies are also needed to explicate the nuance and texture of a group’s lived experience (in this case, displaced Africans currently living in South Africa). In this study, we applied an emic lens to explore how forced African migrants deal with psychological distress stemming from violent, traumatic experiences and precarious resettlement conditions in South Africa. Using a descriptive phenomenological qualitative design, we explored this group’s lived experience of coping with hardship and struggle in tandem with personal strengths and capacities that may help them transcend suffering and support meaning making, growth, and flourishing.

## 2. Methods

### 2.1. Participants

The sample consisted of 14 Africans who were asylees (57.2%) or refugees (42.9%) residing in Durban, South Africa. They were from the Democratic Republic of Congo (DRC; *n* = 5), Burundi (*n* = 4), Eritrea (*n* = 2), Zimbabwe (*n* = 2), and Somalia (*n* = 1). The average age of the participants was 30.27 years (*SD* = 9.3, range = 24–42), and the majority were men (71.4%). In terms of religion, 71.4% self-identified as Christian, and the remainder as Muslim (14.3%) or Rastafarian (7.2%). Half were married (50.0%), and the rest were single (42.9%) or divorced (7.2%).

### 2.2. Procedure

Ethical approval for this study was granted by the Human and Social Sciences Ethics Committee at the University of KwaZulu-Natal. Potential participants were informed about this study’s aim and objectives in a language they understood. They were also informed about the voluntariness of their participation, as well as anonymity, confidentiality, and the use of audio recordings for all interviews. Those who agreed to participate signed a written consent.

Participants were purposively selected to capture diverse lived experiences of local asylees and refugees. Initial contacts with prospective participants were made via the first author’s network of relationships with people and organizations connected to the local asylee and refugee community. Most participants (*n* = 8) were recruited from Refugee Social Services, which is a non-profit organization that offers a range of social services. The remaining participants (*n* = 6) were recruited using snowball sampling via local activist and refugee rights organizations. All interviews were conducted between March and May 2018.

Two of the authors (AT and AMW) developed and refined a semi-structured interview protocol based on a review of the literature. Questions elicited descriptions of participants’ experiences in their own words (e.g., “What were the most stressful experiences you have witnessed or experienced in your home country, during migration, and/or post-migration?”) as well as their coping resources (e.g., “How were/are you dealing with these challenges?”). The first author conducted all interviews in a comfortable location that was convenient for participants to access. Nine interviews were conducted in English, three in Swahili, and two in Tigrigna. A trained interpreter was hired for the Swahili interviews, and Tigrigna is the native language of the first author. Each interview was completed in approximately 60 min. Basic refreshments and transportation costs were provided to participants. Interviews were then transcribed, with those conducted in Swahili and Tigrigna subsequently translated into English.

### 2.3. Qualitative Analysis Plan

This study used a subset of data from a larger project exploring migration experiences, psychological distress, and coping among forced migrants in South Africa [25]. Given our focus on capturing emic perspectives of asylees and refugees, we utilized a descriptive phenomenological data analytic approach. Central to phenomenology is the concept of *epoché*, meaning bracketing or suspension in order to capture a first-person point of view [26]. Bracketing aims to see the phenomena at hand in a new light; thus, the researchers suspend previous knowledge and views underpinning existing attitudes [27]. The analysis was conducted by AT, a post-doctoral research fellow in psychology (African refugee and activist, male), and AMW, a professor of psychology (South African White female). Although it is not possible to achieve complete bracketing, the analytic team took active steps through ongoing discussion and self-reflection to recognize how cultural and educational backgrounds might influence the interpretive process.

A descriptive-phenomenological design is guided by the assumptions of phenomenological reduction and the researchers refraining from drawing on existing knowledge or perceptions [27]. While this analytic method does not lead to theory-building, its strength lies in enabling researchers to retain and center participants’ voices to capture subjective-psychological perspectives [28].

In this study, five steps of data analysis outlined by Giorgi [27] were utilized. First, the analytic team read each transcription to obtain a sense of the whole and become familiar with participants’ lived experiences. The second step involved adopting an open or unspecified attitude (i.e., performing ‘reduction and bracketing’) to explore the phenomenon. Next, the researchers delineated meaning units from within the narratives to capture distinct meanings; this was an iterative process involving discussion among the analytic team until consensus was reached. Fourth, meaning units were transformed into more psychologically sensitive statements using an imaginative variation. This process entailed intuiting and transforming participants’ lived experiences into language, highlighting unique psychological meanings. In the last step, the transformed meaning unit codes were used to describe the psychological structure of participants’ experiences, enabling the researchers to identify meanings shared across participants. Then, thick descriptions highlighting overarching themes were developed and codes with adjacent meanings (e.g., hope and optimism) were grouped together in the write-up.

## 3. Results

Participants shared detailed accounts of their migration experiences, including stressors, psychological difficulties, and coping resources. Concerning stressors, participants described state-sponsored violence in their countries of origin, characterized by the arrest and disappearance of comrades and family members, civil wars, political and tribal persecution, and gross human rights violations [25]. During resettlement in South Africa, participants described problems with securing and maintaining documents, a lack of basic necessities, xenophobic violence, restricted access to public services, insecurity and uncertainty about their livelihoods, communication difficulties, and unemployment.

A variety of mental health difficulties were noted, including PTSD, anxiety, depression, and somatic complaints. Within this context of ongoing challenges and hardships, our interview questions and analytic process were oriented toward exploring character strengths as potential coping resources to promote positive adaptation, PTG, and flourishing. As a result, the following five overarching themes emerged within participants’ narratives: (1) spirituality and religiousness, (2) love and kindness, (3) hope and optimism, (4) persistence and fortitude, and (5) gratitude and thankfulness (see Appendix A). Phenomenological data in each of these categories are detailed below and, to provide context, existing definitions in the literature for each character strength are provided to help frame and contextualize results.

### 3.1. Spirituality and Religiousness

A majority (64.3%) of participants referenced the salience of spirituality and/or religiousness (S/R), encapsulating “the conviction that there is a transcendent (nonphysical) dimension of life” [21]. Etically speaking, spirituality is “an awareness of a being or force that transcends the material aspects of life and gives a deep sense of wholeness or connectedness to the universe” [29]. Though sometimes used synonymously with religiousness, spirituality is oriented toward the subjective experience of connection with what is viewed as sacred, whereas religiousness captures adherence to overarching beliefs and practices associated with a particular faith tradition. Across world religions, connecting with a higher power to endure and transcend suffering is often salient [22]. Despite conceptual distinctions, S/R influence how asylees and refugees make attributions, construct meanings, and engage with their life experience.

From an emic perspective, participants elucidated diverse ways in which S/R supported their migration journey. For example, one participant described how he tried to escape from forced conscription in Eritrea, but was caught by border guards twice, subjected to torture and imprisonment, jailed without a court hearing or sentence, and forced to undergo military training. He reflected on this experience, “My faith in God gives me strength and encouragement, and everything is from God”. A participant from Burundi witnessed the killing of close relatives at the hands of the government, because the armed groups fighting against the government were from his tribe. He and his family fled to Tanzania from ongoing persecution, but he left his parents there and continued to southern Africa because the Burundi government was searching for young people who had fled the country to neighboring countries. This participant drew on his spiritual life to work through internal conflicts about this decision:


*I convinced myself that what I was doing was my decision for my own safety. So, I was praying to arrive safely, and I was hopeful that by God’s will, my family was also doing well, as I was safe and not harmed. I thought that they were also safe. This was giving me a big morale boost and strength.*


Engagement with prayer and trust in God’s providence offered this participant solace and helped regulate difficult thoughts and feelings (e.g., regret, guilt, fear) that arose.

Another participant, who fled the civil war in the DRC, described how her spiritual beliefs offered internal stability and comfort in the face of immense suffering. Having witnessed killings and rape in her town, she fled to Tanzania, where she was tortured by gangs who demanded money. Reflecting on how she survived this experience, she noted, “It is my belief in Jesus that helped me to pass through all the difficulties, because once a person believes in Jesus … the way a believer handles things is different”.

Furthermore, participants shared how their relationships with religious leaders and community helped them to deal with difficulties. One participant recounted how he fled the DRC without saying goodbye to his family when an armed group raided his school and forcefully took young boys to arm them. In South Africa, he could not complete additional schooling due to a lack of finances and has not had stable employment since arriving in the country in 2002. This participant is the sole wage earner in his family, which causes ongoing stress, yet he spoke about how his religiousness guides him:


*The thing that keeps me together is my Christian faith and the support I get from the church and the pastor. My background as a Christian taught me the right manners. My faith helped me not to do bad things. I am encouraged by the Gospel and the Bible.*


This participant’s faith provided comfort and support and helped him avoid risky behaviors (e.g., selling drugs and stolen goods) that he might have turned to as a means of basic subsistence. These examples collectively point to the central role of S/R as a culturally embedded capacity that can promote meaning making and resilience amidst both pre- and post-migration challenges.

### 3.2. Love and Kindness

Over half (57.1%) of the participants’ narratives of dealing with stressors described love and/or kindness. Love involves a disposition toward desiring the good for another person for their own sake, which often translates into attitudes and actions aimed at this end (e.g., approaching others with acceptance, providing support). Love empowers individuals to prioritize the needs of others and make sacrifices for their benefit [30]. Similarly, kindness includes facets such as generosity, nurturance, care, and compassion [21]. Kind people are considerate and unselfish, taking action to support the welfare of others. Both kindness and love can include strong feelings, such as affection and protectiveness.

From an emic perspective, participants’ love for their families, along with the comfort and care received from these loved ones, often evoked a sense of joy and responsibility that helped them persist amidst challenges. One Eritrean participant who left his children in his home country described how their welfare was a primary motivator and source of strength: “I have to stand and overcome challenges to accomplish the expected responsibility for my children”. Many participants cited caring for their family as their greatest concern, with love fueling a sense of responsibility to keep working and striving for a better life amidst exceedingly difficult circumstances. For example, one man from the DRC noted, “Having a family is a support in terms of responsibility. I cannot live my life anyhow. When I look at my wife and two sons, I feel so happy and encouraged to do even more”.

For many, the love and care experienced in family relationships also served as a primary source of comfort, joy, and relief. For example, a female participant from DRC, who witnessed and experienced torture, described how giving birth to her first child was a wellspring of strength:


*I was very happy to see my child alive. It relieves my stress from the traumatic experience. I was very happy because if they had killed me, I was not going to see that child. Then we named the child Joy … she brought joy to our life again.*


A participant from the DRC, who had escaped community persecution and experienced survivor guilt and depression, noted that these symptoms eased when he met his wife two years after arriving in South Africa. He shared that the love they have for each other calmed and helped him begin to envision a future in order to move from surviving toward flourishing:


*I am married and my wife supports me a lot. She is a Christian. We have aimed at something, and we are working together … I think we will achieve it because we are walking together … There is good communication between us. I am seeing really, something changes, a very good life.*


Taken together, participants’ responses elucidated complex interactions between love and kindness with core motivations to persist in the face of forced migration in order to create a better life for themselves and their families.

### 3.3. Hope and Optimism

Nearly half (44.5%) of the participants shared perspectives resonant with hope and/or optimism, described previously in undifferentiated etic language as a “cognitive, emotional, and motivational stance towards the future” [21]. Hope includes a strong belief that, despite the difficulty that one is facing, a better future is possible and achievable [31] and entails one’s capacity to initiate and sustain actions as well as to generate different routes toward intended goals [32]. Similarly, optimism involves expectation or a belief that experiences will be favorable and good in the future [33]. Further, optimism is linked with increased persistence in trying to reach goals and making efforts even in the face of difficulties [34].

From an emic perspective, participants offered embodied and nuanced perspectives on the role of hope amidst forced migration in Africa. One participant, a professional public health worker from Kivu, a war-torn region of the DRC, recounted negotiating with an armed group to gain access to provide care to communities at risk; however, his intentions were misunderstood and his own community burned his property and threatened to kill him for ‘befriending’ the rebels who were persecuting them. Fleeing for his life, he recalled being on the verge of despair when he came to South Africa, but noted the role of hope that motivated him to pursue educational possibilities:


*There was a time I felt my life was ending … I also realized that I did not know what the future would bring. I was working as a car guard. Instead of crying, I decided to add another degree. Then I went back to school. I thought my education would be for nothing if I worked as a car guard. I was motivated to go back to school. Now, I am not doing bad. If I had kept stressing myself, I would not have reached this far.*


This narrative highlights the interplay between forced migrants’ context and both challenges to and opportunities for hope. While a professional with a stable salary in the DRC, following forced migration, this participant experienced a drastic shift in his vocational identity and employment options. In the face of this challenge, engagement with hope supported his positive adjustment and creative adaptation.

For others, optimism was more central, manifesting in how their role as activists could promote community-level changes and contribute to a sense of purpose. Participants reflected on both key aspects and functions of optimism in their lived experience of ongoing uncertainty and adversity. For example, one participant was jailed several times in Zimbabwe due to being part of the opposition leadership and witnessed the killings and disappearance of comrades during the 2005 election. After fleeing to South Africa, he had no stable employment due to limitations imposed by his immigration status. Despite all this, he noted:


*I am always an optimist. In fact, I did another course to upgrade myself. Now I am an activist, a member of an organization that advocates for the rights of refugees and social cohesion among locals and refugee communities. We may not have done big things yet, but we believe there are changes in some people’s minds.*


Another participant, a Zimbabwean who migrated following the country’s economic collapse and was self-employed in South Africa, reflected, “I am building a future … I am not returning back. When I am doing what I am doing, I must be patient and never lose hope. I look upward for ideas and support”. In these examples, optimism is inextricably intertwined with ongoing struggle and barriers, a facet not always accounted for in etic descriptions.

### 3.4. Persistence and Fortitude

Persistence and/or fortitude were evident in some (28.6%) accounts. Etic definitions of these constructs capture continuous and non-flagging efforts toward one’s intended goals, despite failures and adversity [21]. Persistence sustains people and motivates them to keep going in the face of overwhelming obstacles, rather than becoming discouraged and deterred by difficulties or setbacks [35]. Fortitude similarly involves mental and emotional strength to face hardship and adversity with courage.

From an emic perspective, participants shared thick descriptions of how persistence supported their survival and coping. For example, the participant who fled the DRC when an armed group forcefully raided his school described persisting in the face of ongoing setbacks to change his situation of being employed as a car guard in South Africa:


*It was a tough situation, but I was so determined to change my life. I had to do something. Something inside me told me that a car guard was not who I am and what I wanted to be. When I decided to learn, I was told that I am too old to be enrolled in high school. However, I met a pastor who took me to an adult school and paid my school fees. I was bothered about how I would pay rent and buy food if I am going to study as I will not have time for work… Then I had to work as a security guard at night while studying during the day. I was sleeping only a few hours. When I passed the matric [grade 12 national exam], … I convinced myself I could do something.*


When asked about what helped sustain him and fuel his persistence, the participant noted:


*The factors that helped me to pass through my difficulties are first my belief that I always carry with me … no matter the situation I am in, whether good or bad, those things pass, they are not there forever, but for a short time. I live hoping for better.*


This type of narrative is resonant with others, wherein a belief that things will not remain the same forever and that change is possible supports persistence in the face of adversity. Witnessing his efforts to change his circumstances bearing fruits, another participant from the DRC noted, “Things will change… a new day will come and new things as well”.

Another example is a Burundian participant who described fleeing the country after his entire family was murdered. Although he escaped government persecution, he had no one to sponsor his travel or support him when he arrived in South Africa and became homeless. However, giving up was not an option for him, and he embodied fortitude:


*Although I was discouraged, I was also thinking, coming back and making myself ready to try another way to cover my shorts [lack of resources] through different means. That backup feeling was boosting my mind in the situation which weakened my courage and resilience.*


Taken together, these stories exemplify how persistence and fortitude can support resilience and protect against hopelessness and despair among Africans experiencing forced migration.

### 3.5. Gratitude and Thankfulness

Some (21.4%) participants’ narratives also elucidated gratitude and/or thankfulness. Etic understandings of these character strengths focus on acknowledging good things or positive experiences received (i.e., from others, the divine, or the environment), which can often motivate altruistic acts [21]. From this vantage point, gratitude is not limited to saying, “thank you”, but also involves affective and motivational components.

From an emic perspective, participants described the uniqueness of their engagement with gratitude. For example, the Eritrean participant, who had been subjected to torture and imprisonment, was thankful to be alive and see another day. Thinking about peers who were still experiencing violent victimization or had lost their lives, he reflected, “I am thinking about a better tomorrow. I am not discouraged by what happened, and I am trying to take things easy. My past hardship experiences also helped me see things easily.” He went on to describe a sense of deep thankfulness for escaping this oppressive context, suggesting that, emically speaking, gratitude is not only about receiving good things from others but may also be evoked amidst changing circumstances. For multiple participants, gratitude was expressed implicitly for having endured horrific situations, being alive and in a healthy state, lessons learned from past hardship, and new perspectives emerging out of suffering.

On a similar note, a participant from the DRC described loneliness and distrust when he came to South Africa, linking this back to the traumatic impacts of his father’s disappearance. His tribe was collectively accused by the DRC government of assisting tribal rebels, and his father was taken when soldiers raided their home. Fearing he could be next, the participant fled to South Africa. He noted having used alcohol to cope in the past, but cited gratitude for the positive impacts of community involvement in supporting change:


*I was fortunate to run away from the country. My experience in South Africa showed me how things could be changed at home. These made me keep pushing and helped me to discipline myself, even to stop drinking. Even though there are challenges, a better future for everyone is possible. That is the main motto that keeps me going.*


This narrative shows the interconnectedness of character strengths, as the participant’s gratitude for new connections and having escaped from the DRC intersects with a sense of hope and optimism—not only for himself, but also for others. It is noteworthy to mention that such interconnectedness among the thematic areas was evident throughout participants’ narratives.

## 4. Discussion

Drawing from in-depth interviews conducted in multiple languages, our findings contribute to the evolving existential positive psychology literature by exploring African asylees and refugees’ virtue engagement as potential sources of coping and PTG. Our study design and population respond to recent calls for research into “how virtue and well-being constructs intersect with structural inequality, minority stress, and intersectionality”, recognizing that “the diverse ways that virtues are understood, valued, and embodied can vary significantly, often based on the intersections of a person’s culture, S/R beliefs and social location” [17]. Although an abundance of research on character strengths has been conducted in Western, educated, industrialized, rich, and democratic (WEIRD) samples [36], less is known about the interplay of religion and culture with such capacities in diverse and often underrepresented contexts. A central thread across study narratives was the journey of transcending victimhood and beginning to see themselves as survivors who could work to change their circumstances, no matter how challenging. Consistent with dual-factor models of mental health that attend to both suffering and flourishing [16], participants described (a) pre- and post-migration stressors that fuel psychological distress as intertwined with (b) cogent examples of how character strengths supported their adaptive coping, facilitated perspective taking, and oriented them toward possibilities for flourishing amidst adversity.

Rather than importing etic understandings of these character strengths, which are largely Eurocentric, our data centered forced migrants’ unique perspectives and vantage points in a South African context. Participants often referenced complex existential dilemmas—such as losing and leaving family behind, losing one’s livelihood, and the awareness of friends who could not escape and were still being tortured—as they spoke about how character strengths buffered some of the distress of these life experiences and helped them hold complexity and uncertainty. For this population who experienced protracted hardship and disenfranchisement, character strengths did not function by helping them to avoid difficult emotions or “just think positive”, what Captari et al. [17] formulated as virtue bypass. Rather, drawing on these internal and communal capacities bolstered participants’ sense of self and will to live and survive. Furthermore, while character strengths are often conceptualized as individual-level phenomena in the West, our findings elucidate these strengths as emerging within—and at times inseparable from—participants’ experiences in close relationships (e.g., romantic partners, children, parents, and other family members) and their faith community (e.g., pastor, church). This aligns with third-wave positive psychology’s attention to sociocultural and systemic impacts on individual well-being [17].

Within this sample of African refugees in South Africa, S/R emerged as the most frequently referenced capacity. Theodicies of suffering and belief in ‘God’s will’ as a greater force to explain what is beyond human control helped participants construct a sense of coherence out of tragedy and begin to metabolize the psychological impacts of trauma. Spiritual explanations for survival provided many participants with the strength to face past and current difficulties. Given the complexity of surviving atrocities and migrating successfully, some participants described survivor guilt after family members and friends were imprisoned or killed in their country of origin. S/R seemed especially salient in these situations by offering attributions other than self-blame; for example, the sense that God was present with them, they survived by divine providence, and acceptance of the mystery of suffering and loss. Salient spiritual meanings frequently included faith in a divine purpose, which served as a source of comfort and strength, down-regulating trauma-related affects. At times, participants wondered aloud how they dealt with overwhelming stressors (e.g., lack of income), especially when there was no foreseeable solution, yet their attributions such as “God has His own ways” and “the Spirit of God is leading us” suggest an active process of surrender, including acknowledgment of what cannot be understood, and partnering with God. These S/R processes are resonant with similar research in other populations [37].

Study participants also frequently referenced God, Bible teachings, and the influence of the faith community as sources of direction and encouragement. This affirms previous scholarly work on S/R as potentially restoring (a) agency and purpose amidst heightened vulnerability and (b) a sense of worth and being cared for by God in the face of dehumanization [38]. Previous research among asylees and refugees has found religious coping (e.g., prayer, spiritual practices, trusting God, connection with one’s faith community) to help in managing daily stressors [39], buffering the impacts of traumatic stress and acculturation challenges [40], and supporting PTG [4]. Specifically, evidence to date suggests that individuals who lean into and strengthen their S/R commitments may be more likely to experience PTG [39]. In this regard, the findings of this study might be relevant to the empirical question of what factors promote PTG. Previous studies have emphasized the noteworthy influence of S/R and character strengths in fostering PTG [41,42]. For instance, a review by Chan et al. [41] demonstrated that social support, spirituality, and optimism were associated with PTG. Similarly, Ersahin’s [42] study conducted among Syrian refugees revealed that problem-focused coping and religiosity were significantly associated with PTG. This may be due to how S/R can be a powerful resource in facilitating a sense of meaning and coherence, as well as cultivating positive relationships and a sense of community [43,44]. Additionally, engaging in S/R can be a pathway to seeking comfort and closeness to a higher power, gaining a sense of control and/or perspective, and/or facilitating personal transformation [45,46].

It is noteworthy that spiritual struggles (e.g., anger toward God) were not overtly described by participants, although interview questions were oriented more toward strengths and thus not designed to explore all facets of S/R. Given empirical evidence for such struggles as occurring amongst asylees and refugees [47], and the potential negative implications of spiritual struggles for mental health and well-being [48], this is an important area of future investigation.

Previous empirical work has examined S/R among asylees and refugees; however, novel to this study is the examination of the interplay between S/R and other strengths—and consideration of how this may impact coping, mental health, and relationships. First and foremost, love and kindness promoted adaptive coping by fostering forced migrants’ sense of responsibility, sacrifice, and protectiveness. These sources motivated participants (a majority of whom were male) to act in ways that transcended the self, such as prioritizing their families’ welfare and needs as well as working hard to meet responsibilities (e.g., as wage earners, parents). Particularly among the fathers in the sample, fulfilling the culturally expected roles of protection and provision motivated and empowered them to keep fighting for survival, as giving up would be failing their responsibility as a partner and parent. Family relationships—and particularly a partner’s love—were also described as sources of relief, comfort, and satisfaction. Furthermore, social connections with others who had similar experiences of displacement were described as salient contexts to both give and receive care and support. Although empirical work has previously linked social and family support with well-being and PTG in these populations [3], identification of (a) underlying character strengths, (b) the motivating psychic function of parenthood, and (c) reciprocal influences between self and family warrant further investigation.

Hope and optimism, along with other character strengths described by participants, were narratively linked with S/R as an undergirding meaning system. This is resonant with recent scholarly work noting synergy between S/R and character strengths, such that “spirituality is vitally concerned with promoting character strengths … [and] character strengths can enhance and deepen spiritual practices, rituals and experiences” [49]. Participants’ engagement with hope stemmed from a broader S/R philosophy of life viewing circumstances as changing—rather than static—life events. Thus, their mental attitude of anticipation was accompanied by action and a belief that they could work to change their situation (e.g., from car guard to academy enrollment). Such anticipation and action were fueled not merely by willpower or belief in human possibility; rather, the deeply held sense that God was with them and would take care of them fostered divine trust and a sense of partnership with God. Although this finding aligns broadly with Callina et al.’s [50] triadic formulation of hope (i.e., positive future expectations, agency, trust), African refugees’ lived experiences were distinct in that their sense of trust was in a divine, all-powerful being, not merely themselves, with the expectation to restore a sense of control by proxy (e.g., God is in control, and I trust Him). Furthermore, participants’ optimism accords with the notion of tragic optimism, wherein one does not refuse the reality of darkness in life but searches for meaning despite tragic experiences [51].

The intergenerational transmission of trauma impacts has been well-documented among forced migrants and refugees [52], and potential mechanisms of transmission are being increasingly elucidated, including “insecure attachment, maladaptive parenting styles, diminished parental emotional availability, decreased family functioning, accumulation of family stressors, dysfunctional intra-family communication styles, and severity of parental symptomology” [53]. When it comes to considering the roles of S/R and related strengths in this population, orienting beyond just the individual is vital. Researchers and practitioners would do well to consider how engagement with such virtues within family and kinship networks might support an upward spiral of well-being and intervene to help buffer against the intergenerational transmission of trauma through altering some of the afore-mentioned mechanisms. Our findings suggest that close relationships seem to hold particular potential for both embodying *and* experiencing virtues, which appear to play influential roles in adaptive coping, meaning making, and growth. While some work has been conducted bringing a family systems perspective to PTG [54], this may be especially important in collectivistic cultural contexts.

To date, research with asylee and refugee populations has mainly examined one or two character strengths at a time [55]. Although this is a valuable start, narratives of the asylees and refugees in our sample point toward the need for greater complexity in formulations of virtue engagement. For example, participants’ accounts of persistence and fortitude are exemplary not only in the sheer level of torture, violence, and daily hardships endured, but also in how other strengths like love, hope, and gratitude seemed to be critical ingredients in fueling persistence and fortitude. Furthermore, how these strengths unfold and are embodied within close relationships—and the impact on those relationships (e.g., the participants’ spouses, children)—is an important empirical question. From a phenomenological descriptive vantage point, our focus was not on theory building or constructing an overarching framework; however, the character strengths identified herein may operate much like an interconnected web that offers internal stabilization, fosters adaptive coping, and facilitates PTG. Bi-directional relationships and spillover effects between these capacities are worthy avenues for future exploration and are important when considering clinical and community interventions.

### 4.1. Clinical and Community Applications

In exploring strengths and capacities, it is vital to not lose sight of the chronic mental health burden in asylee and refugee populations. We encourage mental healthcare providers and community leaders to utilize dual-factor understandings of mental and spiritual health to better elucidate the lived experiences of forced migrants, recognizing that emotional and psychological suffering exist alongside the character strengths mobilized to seek well-being and growth. These domains of human experience are distinct and interconnected, and equal attention to these dark and bright sides is needed for both assessment and intervention. Neglecting either side fails to capture the complexity evident in the present study’s narratives. Culturally responsive treatment involves a holistic focus and being careful not to reduce a person to their symptoms; at the same time, mental health conditions must be taken seriously and attended to accordingly. There may be a subset of individuals whose level of functioning is compromised to the point that experiencing the well-being benefits of virtue engagement may be less feasible until stabilization occurs.

Considering the intergenerational transmission and impacts of unresolved trauma, practitioners should be mindful of adverse events experienced in previous generations, which, if not processed, may continue to negatively impact the well-being of second and third generations. A systemic perspective to mental health support and intervention can orient practitioners to this complexity. As part of this, attention to the intergenerational transmission of strengths [56] is a fascinating application of this research. Those experiencing forced migration may well identify the character strengths they draw from as stemming not only from themselves, but perhaps also as a legacy passed down from previous generations who also endured and transcended suffering.

In the context of South Africa, where an estimated 75% of people who are living with a mental health condition do not receive formal treatment, access to mental health services for refugees and asylees is further constrained by precarious socioeconomic conditions and, sometimes, by cultural and spiritual beliefs. Nevertheless, sensitivity is needed when attempting to integrate character strengths into psychotherapy with these populations. Such capacities may not be most effectively facilitated through verbal intervention, particularly early in treatment. For example, it may not be particularly helpful to tell an asylee to ‘have hope’ or ‘be grateful’, which could be experienced as misattuned and minimizing their distress. Rather, in line with best practice guidelines for trauma-informed work, inviting narrative reflection about experiences of suffering and hardship is likely to open up possibilities for inquiry into strengths and capacities (e.g., ‘What is it that has helped you survive? What do you turn to for strength and comfort amidst all this?’). Existential positive psychology recognizes that “suffering is an undeniable, self-evident reality” that cannot be ignored or avoided [57]. Whether in psychotherapy or other settings (e.g., faith communities, schools), interventions that (a) validate the legitimacy of survivors’ suffering (acknowledging both pre- and post-migration stressors) and support meaning-making processes and (b) invite survivors to acknowledge their strengths and look for moments of flourishing, are most likely to be facilitative of PTG.

### 4.2. Limitations and Future Directions

This study had several methodological limitations. Although participants were diverse with regard to country of origin, a majority of the sample was male and self-identified as Christian. Our findings are unlikely to capture the full spectrum of forced migration experiences among Africans, particularly diverse religious groups, vulnerable populations (e.g., children), and those who migrated to other African countries that differ from South Africa in their humanitarian, political, and socioeconomic infrastructure. Future research among African asylees and refugees should seek to capture more diverse lived experiences across faith traditions, genders, sexual orientations, hosting society, and time since migration. Interviewing multiple family members could also elucidate important generational differences in experiences and perspectives, as well as explicate the potential of character strengths to help buffer intergenerational trauma impacts. Furthermore, because the current study was part of a larger investigation, in-depth understandings of each of the character strengths identified were beyond its scope. Important future areas of empirical exploration concerning the role of character strengths among Africans (and others) experiencing forced migration could include inquiries into (a) the interplay between these character strengths, recognizing the interconnectedness evident in this sample, (b) situations in which these capacities do—and do not—support asylee and refugee well-being, in line with concepts of virtue bypass and burdened virtues [58] and (c) how intersectionality of identities (e.g., identifying as a woman, LGBTQ, and/or with a non-majority religion) may complicate coping, attributional, and PTG processes.

## 5. Conclusions

The character strengths emerging as sources of coping in this study have been studied across cultures, but only more recently been examined in refugee populations. Notable aspects of this study worthy of highlighting are its emphasis on the positive impact of S/R on PTG and overall well-being. In this sample, engaging with S/R activities enabled individuals experiencing forced migration to find and make meaning, achieve acceptance, and seek solace and connection with a higher power. Narratives of S/R were frequently interconnected with participants’ ability to draw from character strengths in a deep and meaningful way (e.g., spiritual beliefs that fuel optimism and hope). Moreover, relationships were highlighted as a frequent context for experiencing *and* embodying virtues; for example, love and kindness from family members bolstered commitment to not give up when faced with seemingly insurmountable challenges, and participants’ embodiment of strengths, such as hope, gratitude, and fortitude, have the potential to catalyze growth and healing on a family level. Against the backdrop of growing public health concerns over forced migration in Africa, we hope that our findings spark increased interest in emic perspectives on virtue engagement in the African context and elucidate the potential for existential and systemic positive psychology frameworks to meaningfully capture the complexity of this population’s lived experiences.

## Data Availability

The coded data can be available on request due to ethical restrictions.

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
