# Peer review of "Coping Resources among Forced Migrants in South Africa: Exploring the Role of Character Strengths in Coping, Adjustment, and Flourishing"

_ijerph, 2023, doi:10.3390/ijerph21010050_

Round 1

Reviewer 1 Report

Comments and Suggestions for Authors

Thank you for the opportunity to review this paper. It reads very well and introduces a different view on refugees and migrants in vulnerable conditions. The paper is theoretically informed by PTG, and I would advise that there is a more thorough discussion of PTG and how it has been used in previous studies. The paper must also be clearer on its contribution/s be it empirical and/or theoretical. With these revisions, I believe you have a very good paper. All the best in progressing it further.

Reviewer 2 Report

Comments and Suggestions for Authors

Important study, well written. I appreciate the clarity and method transparency.

Typo on line 61 - South Africa

Use of comma needs to be revised throughout - sometimes Oxford comma, sometimes not. There are many situations where a sentence can use a comma to split its parts into more digestible chunks. For example, line 106, add comma after “hopeful moments, as well as…” 

The framework of the study, it will be useful to consult Joy DeGruy’s “Post Traumatic Slave Syndrome” book. While focused on the US context and legacy of slavery, it has significant contribution to the understanding of trauma and its becoming embedded into culture across generations; Resmaa Menakem's “My Grandmother’s Hands” also examines generational trauma.

I was interested in seeing a bit more connection between the strategies for coping with trauma identified in the study and the connection to others (beyond the obvious love for others / family, for instance). How does their spirituality / faith / religion serve as a connector to community - old or new in the refugee space? This can connect with the practical implications, for instance, to support an environment where one can practice their spirituality / faith / religion alone or in a group.

I look forward to seeing this in print.

Reviewer 3 Report

Comments and Suggestions for Authors

I would like to thank authors for their efforts to look into the subject matter of coping resources associated with character. While the sample size is small, it is noted that qualitative research is designed to describe a phenomenon, not to validate it. The article offers an solid overview of the methodology, research methods, and findings with an analysis. The draft is well written, the main argument is clearly presented, supported with the evidence, and discussed. The inclusion of researcher's observations represents a plus. The draft acknowledges, albeit indirectly, the potential biases contained in the research findings, which are inherent to the data collection method (stories of the migrants themselves). The draft complies with standards for accuracy in references and definitions. The conclusions read somewhat simplistic and maybe implicit in advance, which make qualitative aspect of research shallow. I would recommend to expand along the lines already demarcated in the draft.

Introduction:

29-30: a suggestion to add the numbers.

Materials and Methods: the sample size, as the authors point it, is limited, but in line with the methodology, and selection criteria for choosing the sample is justified and described with a sufficient level of detail.

Analysis: the tools deployed for the analysis are well described.

Results: clear and detailed overview of results strengthens the draft.

Conclusion section could be expanded.

I wish the authors success in continuing their research work.
